# Major Adverse Cardiovascular Events and Mortality Prediction by Circulating GDF-15 in Patients with Type 2 Diabetes: A Systematic Review and Meta-Analysis

**DOI:** 10.3390/biom12070934

**Published:** 2022-07-04

**Authors:** Suyi Xie, Qi Li, Andrea O. Y. Luk, Hui-Yao Lan, Paul K. S. Chan, Antoni Bayés-Genís, Francis K. L. Chan, Erik Fung

**Affiliations:** 1Department of Medicine and Therapeutics, The Chinese University of Hong Kong, Prince of Wales Hospital, Hong Kong, China; suyixie@cuhk.edu.hk (S.X.); qlizzy@hotmail.com (Q.L.); andrealuk@cuhk.edu.hk (A.O.Y.L.); hylan@cuhk.edu.hk (H.-Y.L.); fklchan@cuhk.edu.hk (F.K.L.C.); 2Laboratory for Heart Failure + Circulation Research, Li Ka Shing Institute of Health Sciences, and Gerald Choa Cardiac Research Centre, Faculty of Medicine, The Chinese University of Hong Kong, Hong Kong, China; 3Heart Center of Henan Provincial People’s Hospital, Department of Cardiology of Central China Fuwai Hospital, Henan Key Laboratory for Coronary Heart Disease Prevention and Control, Central China Fuwai Hospital of Zhengzhou University, Zhengzhou 450003, China; 4Hong Kong Institute of Diabetes and Obesity, The Chinese University of Hong Kong, Hong Kong, China; 5Li Ka Shing Institute of Health Sciences, The Chinese University of Hong Kong, Prince of Wales Hospital, Hong Kong, China; 6CARE Programme, Lui Che Woo Institute of Innovative Medicine, Faculty of Medicine, The Chinese University of Hong Kong, Hong Kong, China; 7Department of Microbiology, Faculty of Medicine, The Chinese University of Hong Kong, Hong Kong, China; paulkschan@cuhk.edu.hk; 8Stanley Ho Centre for Emerging Infectious Diseases, Faculty of Medicine, The Chinese University of Hong Kong, Hong Kong, China; 9iCor, Hospital Universitari Germans Trias i Pujol de Badalona, 08916 Badalona, Spain; abayesgenis@gmail.com; 10ICREC Research Program, Germans Trias i Pujol Health Science Research Institute, Can Ruti Campus, 08916 Badalona, Spain; 11Department of Medicine, Universitat Autònoma de Barcelona, 08916 Barcelona, Spain; 12CIBER Cardiovascular, Instituto de Salud Carlos III, 28029 Madrid, Spain; 13Centre for Gut Microbiota Research, Faculty of Medicine, The Chinese University of Hong Kong, Hong Kong, China; 14Department of Epidemiology and Biostatistics, School of Public Health, Faculty of Medicine, St Mary’s Campus, Imperial College London, London W2 1PG, UK

**Keywords:** growth differentiation factor 15, adverse cardiovascular outcomes, coronary artery disease, heart failure, type 2 diabetes, mortality

## Abstract

Background: Growth differentiation factor 15 (GDF-15) is a homeostatic cytokine that regulates neural and cardio-metabolic functions, and its release is increased in response to stress, injury, and inflammation. In patients with coronary artery disease and heart failure (HF), three separate meta-analyses have found that elevated circulating GDF-15 was predictive of major adverse cardiovascular events (MACE), but none has evaluated its effects on incident MACE including HF and mortality hazard in type 2 diabetes. Methods: MEDLINE, EMBASE, and Scopus databases were queried. Articles that met the predefined eligibility criteria, including prospective studies that reported adjusted hazard ratios (aHRs), were selected according to the Cochrane Handbook and PRISMA guidelines. Study endpoints were (1) MACE including HF, and (2) all-cause mortality. Different GDF-15 concentration measurements were harmonized using a validated mathematical approach to express log_2_-transformed values in per standard deviation (SD). Study heterogeneity (*I*^2^), quality, and bias were assessed. Results: 19354 patients in 8 prospective studies were included. In 7 studies that reported 4247 MACE among 19200 participants, the incident rate was 22.1% during a median follow-up of 5.6 years. It was found that four of eight studies included HF decompensation or hospitalization as a component of MACE. In 5 studies that reported all-cause mortality, 1893 of 13223 patients died, at an incidence rate of 15.1% over 5.0 years. Of note, each 1 SD increase of log_2_[GDF-15] was associated with aHRs of 1.12 (1.09–1.15, *I*^2^ = 5%, *p* < 0.000001) and 1.27 (1.11–1.46, *I*^2^ = 86%, *p* = 0.00062) and for MACE and all-cause mortality, respectively. Conclusion: Elevated circulating level of GDF-15 was robustly predictive of MACE in patients with T2D but its prognostic significance in the prediction of mortality requires further studies.

## 1. Introduction

Growth differentiation factor 15 (GDF-15), or macrophage inhibitory cytokine 1 (MIC-1), is a 308-amino acid protein encoded by a 1.2-kilobase transcript of the *GDF15* gene located on chromosome 19p13.11. Following post-translational processing, GDF-15 is released as a 25-kDa dimer consisting of 224 amino acids that signals through its receptor, glial-derived neurotrophic factor receptor alpha-like (GFRAL), and co-receptor, Ret [1]. Whereas GFRAL is prominently expressed in the brain (particularly, in the area postrema and the nucleus tractus solitarius for appetite regulation) and at lower levels in other tissues including adipose tissue in the human, Ret is widely expressed and may signal in complex with cell-surface or soluble GDF-15/GFRAL [1]. GDF-15 originates from a wide range of cells (e.g. adipocytes, cardiomyocytes, vascular smooth muscle cells, endothelial cells) and tissue types, and is released in response to tissue injury, cellular stress, and inflammation [2]. In addition to playing important regulatory roles in energy metabolism, body weight, appetite, and immune response, GDF-15 is elevated in cardiometabolic diseases, including hypertension [3,4], diabetes mellitus [5,6], coronary heart disease [7], and heart failure (HF) [8,9]. While GDF-15 has antihypertrophic effects on cardiac remodeling and counter-regulates inflammation, studies have shown that chronic elevation of GDF-15 can result in anorexia, inhibition of muscle growth, weight loss, and cachexia [1].

Previously, GDF-15 has been examined as a biomarker to indicate CVD risk and all-cause mortality in three meta-analyses of patients with HF and/or acute coronary syndrome, irrespective of diabetes status [10,11,12]. However, those meta-analyses were limited by a lack of clarity on the definition of study endpoints, harmonization of GDF-15 levels, confounder adjustment, and/or characteristics of the study population [10,11,12]. In view of conflicting findings from separate studies [13,14], and a knowledge gap in an important patient population of type 2 diabetes that has not been closely examined, we performed a time-to-event meta-analysis to determine the impact of elevated GDF-15 levels on incident major adverse cardiovascular events (MACE) [15] including HF, and all-cause mortality, and demonstrated the successful application of a validated mathematical approach to harmonize HRs for the different measurements of GDF-15 concentration.

## 2. Materials and Methods

### 2.1. Data Sources and Search Strategy 

Articles were searched on MEDLINE, EMBASE, and Scopus from inception of the databases until December 2021. Guidance and recommendations provided in the Cochrane Handbook of Systematic Reviews (http://handbook.cochrane.org (accessed on 1 May 2022)) and the Preferred Reporting Items for Systematic reviews and Meta-Analyses Statement (PRISMA) were followed. The PRISMA checklist was used in carrying out this study. 

### 2.2. Eligibility Criteria

The following search terms were used: (GDF-15 OR MIC-1) AND (diabetes mellitus OR diabetes OR prediabetes OR hyperglycemia OR glucose OR insulin resistance). Two reviewers (S.X., Q.L.) independently evaluated the studies’ eligibility by screening literature titles and abstracts. Original research articles meeting the eligibility criteria were retrieved and reviewed. The inclusion criteria were: (i) type 2 diabetes; (ii) investigations into how circulating GDF-15 predicted risks of cardiovascular disease or mortality, (iii) human individuals, (iv) subjects aged 18 years or above, and (v) articles published in English. The exclusion criteria were: (i) basic science or animal research, (ii) subjects younger than 18 years old, (iii) type 1 diabetes, (iv) pregnant women, and (v) not primary and original research articles including reviews, commentaries, editorials, conference abstracts, or letters. 

### 2.3. Definition of Study Endpoints

The study endpoints were (1) MACE, a composite of non-fatal MI, stroke, CV death, and/or HF events (hospitalizations, initiation of loop diuretics, or NT-proBNP elevation) and/or revascularization and/or worsening arrhythmia, critical limb ischemia, or venous thromboembolism; and (2) all-cause mortality.

### 2.4. Data Extraction

The characteristics, outcomes definitions, and adjusted confounders of the included studies were documented by two independent reviewers for descriptive analysis. Disagreement was resolved through discussion with and adjudication by a third reviewer. Data on the most fully adjusted hazard ratios (HRs) and 95% confidence intervals (Cis) were extracted. We extracted HRs from the most fully adjusted models in the respective study, or the models that consisted of the most common covariates among the included studies.

### 2.5. Data Harmonization and Statistical Analysis

Median or mean values of blood GDF-15 concentrations were extracted from studies, in respect of the distribution of the study populations. All included studies used log-transformed concentrations, log(GDF-15) or log_2_(GDF-15), in the respective Cox regression models. We standardized the HRs in the same units and pooled the HRs in per SD of log_2_(GDF-15). Studies that reported HRs in the unit of per IQR log(GDF-15), per log(GDF-15), per log_2_(GDF-15), or per SD log(GDF-15) were converted to per SD of log_2_(GDF-15) according to sample sizes and the dispersion characteristics of GDF-15 levels [16,17,18,19] (Appendix A).

HRs were converted using three equations as shown below. Fixed ratio between log(x) and log_2_(x):(1)$$\frac{\log\left(x\right)}{\log_{2}\left(x\right)}=\frac{\frac{\log_{k}\left(x\right)}{\log_{k}e}}{\frac{\log_{k}\left(x\right)}{\log_{k}2}\;}\;=\frac{\log_{k}2}{\log_{k}e}=\mathrm{loge}2\;$$

Estimating *HR*(*x*) for *n* unit change of *x*:(2)exp(coef(x) × n)=exp(log(HR) × n) 

Estimating SD from quantiles by the method of Wan et al. [17]:(3)$${\mathrm{SD}}_{\mathrm{est}}\approx \frac{q3\;-\;q1}{2\Phi^{-1}((0.75n\;-\;0.125)/n+0.25))}$$
where SD_est_ was defined as the estimated standard deviation; *n*, number; *q*, quantile; *Φ*^−1^(*z*) was the upper *z*th percentile of the standard normal distribution.

Results from eligible studies were pooled and meta-analyzed using the random-effects model with inverse-variance weighting and visually displayed by forest plots. Potential covariates were analyzed for effects on GDF-15 levels in predicting the composite CVD outcomes. The presence (Figure 2A and Figure 3A) or absence (Figure 2B and Figure 3B) of canagliflozin treatment in the study of Sen et al. [20] and its effects on the overall pooled estimates were assessed. A 2-tailed *p*-value of < 0.05 was considered statistically significant.

Study heterogeneity was assessed by the total heterogeneity/total variability (*I*^2^). *I*^2^ threshold values of 25%, 50%, and 75% were regarded as low, moderate, and high, respectively. Review Manager 5.3 (The Cochrane Collaboration) and R 4.0.3 (R Foundation, Vienna, Austria) software packages were used. 

### 2.6. Assessment of Publication Bias and Study Quality

Funnel plot analysis and Duval-Tweedie’s trim and fill test were used to assess publication bias [21]. Quality of studies was individually assessed using the Newcastle-Ottawa scale [22], and verified (Appendix A).

## 3. Results

### 3.1. Characteristics of the Included Prospective Studies

The workflow of our search strategy and selection of articles was summarized (Figure 1). A total of 19543 patients with type 2 diabetes from prospective studies that reported hazard ratios [13,14,20,23,24,25,26,27], including one study with 1561 individuals who had undiagnosed diabetes and confirmed dysglycemia [24], were entered into the meta-analysis (Table 1). The median duration of type 2 diabetes was 8.7 years, as reported in 4 of 8 studies only. Hypertension was present in 66.9% to 83.3% of patients. Other comorbidities are shown in Table 1. The median duration of follow-up in eight studies was 5.0 years. Table 2 summarized the definitions of MACE and adjusted confounders in each included study.

### 3.2. Elevated GDF-15 and Risks of Future MACE

Seven prospective studies including 19200 participants and 4247 composite endpoints were analyzed for MACE. For each SD increase in log_2_(GDF-15), the hazard for MACE was significantly increased (pooled adjusted HR 1.21, 95% CI 1.11–1.33, *p* < 0.0001, *I*^2^ = 87%) (Figure 2A). Sensitivity analysis identified Pavo et al. [26] and study subjects who received canagliflozin in the study of Sen et al. [20] as major sources of study heterogeneity. When both groups were removed, the study heterogeneity markedly improved (*I*^2^ = 5%), while the hazard estimate remained statistically significant (pooled adjusted HR 1.12, 95% CI 1.09–1.15, *p* < 0.00001) (Figure 2B).

### 3.3. Elevated GDF-15 and Hazard of All-Cause Mortality

In the five prospective studies reporting mortality that were included, 1893 all-cause deaths occurred in 13223 patients with type 2 diabetes. For each 1 SD increase in log_2_(GDF-15), the hazard estimate was significantly increased (pooled adjusted HR 1.47, 95% CI 1.23–1.75, *p* < 0.0001, *I*^2^ = 95%) (Figure 3A). When both Pavo et al. [25] and study subjects who received canagliflozin in the study of Sen et al. [20] were excluded, the pooled adjusted HR decreased (1.27, 95% CI 1.11–1.46, *p* = 0.0006), while study heterogeneity marginally improved (*I*^2^ = 86%) (Figure 3B).

### 3.4. Ascertainment of Quality of Study and Publicaiton Bias

The relatively high quality of the included studies were ascertained (Appendix A). Initially, the funnel plot of GDF-15 in predicting MACE appeared asymmetric and indicated potential publication bias (Appendix A). However, regardless of the presence or absence of three studies identified as potential causes of asymmetry in the trim-and-fill analysis, the resulting effect size was similar (1.11, 95% CI 1.06–1.16) as the pooled effect size in our meta-analysis (Figure 2). Both Begg’s test (*p* = 0.19) and Eggers’ test (*p* = 0.15) did not detect the presence of statistically significant asymmetry, suggesting that the pooled effect size of GDF-15 in predicting MACE was reliable.

## 4. Discussion

To our knowledge, this was the first time-to-event meta-analysis on prospective studies that summarized in patients with type 2 diabetes the effects (hazards) of elevated GDF-15 in predicting incident MACE including HF, and all-cause mortality, during a median follow-up of 5.6 years. Prior to this, no meta-analysis has specifically focused on this important patient population. Our meta-analysis has applied previously validated mathematical equations [17,18,19] to harmonize GDF-15 metrics across different studies to derive hazard estimates (see Methods). Our main findings were that each 1 SD increment in log_2_-transformed GDF-15 concentration was associated with (1) a 12–21% increase in the risks for future MACE (Figure 2), and (2) a 27–47% increase in the hazard of all-cause mortality (Figure 3), depending on stringency of study inclusion and heterogeneity. These findings were similar to those reported in the Framingham Heart Study that each SD increment of GDF-15 level was associated with a 13% increased risk for future MACE in individuals with diabetes [28].

The effects of circulating GDF-15 on adverse outcomes in patients with HF or coronary artery disease have been meta-analyzed in three studies previously [10,11,12], but diabetic patients were a minority in each of those studies and none has specifically summarized the effects of elevated GDF-15 in patients with diabetes. Given that GDF-15 level is increased in not only patients with vascular inflammation, coronary artery disease, and heart failure [11,12], it has been unclear if the totality of evidence in the literature concur that patients with type 2 diabetes were similarly affected. 

The recent discoveries and reinterpretations of the complex functional roles of GDF-15 beyond coronary artery disease and HF [1], particularly diabetes [29,30,31] and cardiometabolic conditions [29,30,31], have led us to conduct this meta-analysis focusing on patients with type 2 diabetes. Although increased circulating GDF-15 levels have been closely associated with cardiometabolic disorder and cardiovascular disease [1,32,33], its elevation may also reflect insulin responsiveness [34] and the therapeutic effects of antidiabetic medications [35,36]. As such, GDF-15 serves not only as an indicator of dysmetabolism but exerts also functional cardiometabolic effects. In this study, we also attempted to clarify the effects of canagliflozin, an SGLT2 inhibitor, on GDF-15 levels and the study outcomes [20]. We compared the inclusion (Figure 2A and Figure 3A) and removal of patients who had received canagliflozin in CANVAS (Figure 2B and Figure 3B), a placebo-controlled randomized clinical trial, but observed no significant difference in the adjusted HRs for MACE (1.21 [1.11–1.33] and 1.22 [1.11–1.34], respectively), or all-cause mortality (1.47 [1.23–1.75] and 1.49 [1.22–1.81], respectively). Future prospective studies are needed to ascertain the effects of this important cardiometabolic drug class on incident MACE including HF, and mortality.

Apart from its role as a biomarker, the genetic and quantitative association of GDF-15 with the etiology of atherosclerotic and metabolic CVD is complex and controversial. GDF-15 levels are determined by genetic and other complex lifestyle-related factors (e.g., smoking, diet, physical activity) [37]. Although data on this topic are scarce, several studies have found that single nucleotide polymorphisms (SNPs) in the *GDF15* gene could partly explain circulating levels of GDF-15 [37,38]. Indeed, a meta-analysis summarized that nine SNPs in *GDF15* explained approximately 21% of the variance in blood GDF-15 concentration [38]. However, a recent Mendelian randomization study in 2.6 million individuals from 5 genome-wide association studies found no evidence of causality between *GDF15* SNPs and the incidence of stroke, HF, or nonischemic cardiomyopathy [39]. Further mechanistic studies are needed to fully characterize how the potent functionalities of GDF-15 could be harnessed, if possible, for improving cardiometabolic and vascular health.

There are several limitations in our study. First, we could not obtain individual-level data from authors of the original articles to examine the effects of variables including antidiabetic drugs. An individual-level meta-analysis could further improve the precision of our estimates by analyzing effects of individual variables or potential confounders, and potentially allow for an in-depth analysis of the effects of antidiabetic medications. Second, the study of Gerstein et al. included 1561 individuals without “previous diabetes mellitus” but had impaired glucose tolerance, impaired fasting glycemia and evidence of prediabetes. These individuals were included along with 6840 patients with established type 2 diabetes [24] and accounted for approximately 8.1 % [1561 of 19200] and 11.8% [1561 of 13223] of patients in the meta-analysis for incident MACE and all-cause mortality, respectively. It is possible that that might have potentially reduced the magnitude of the effect size and precision of our estimates. Third, while the removal of two major sources of heterogeneity [20,25] led to an significant improvement in the outcome of incident MACE (Figure 2), we were unable to identify the source of high study heterogeneity for all-cause mortality (Figure 3).

## 5. Conclusions

This meta-analysis found compelling evidence that elevated circulating GDF-15 level is associated with increased risk for incident MACE including HF in patients with type 2 diabetes. While there is suggestive evidence that elevated GDF-15 is indicative of increased mortality hazard, the quality of those data are insufficient for a conclusive interpretation.

## Figures and Tables

**Figure 1 biomolecules-12-00934-f001:**
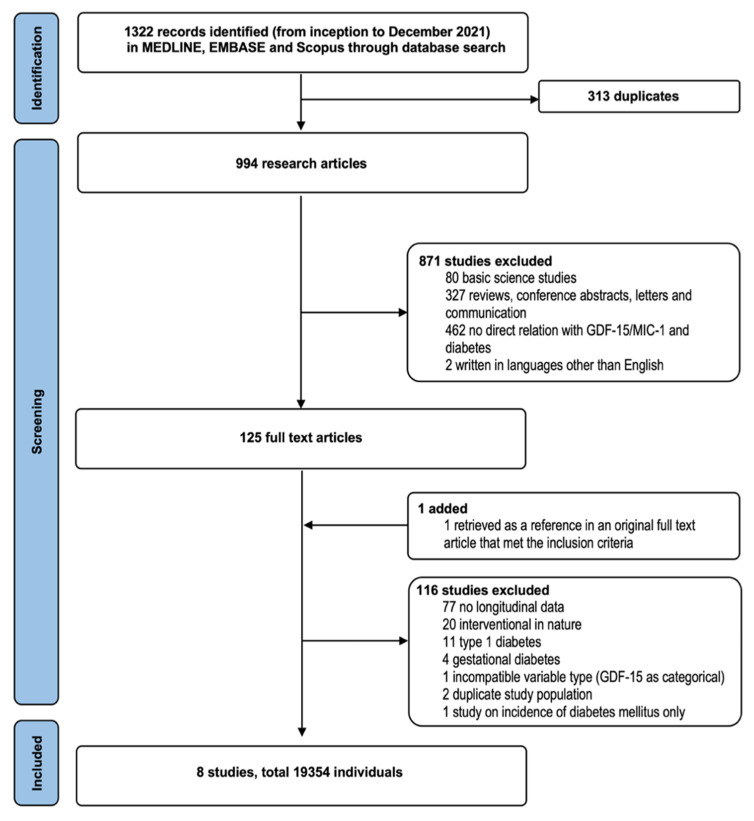
PRISMA flow diagram showing the literature search and article selection strategy.

**Figure 2 biomolecules-12-00934-f002:**
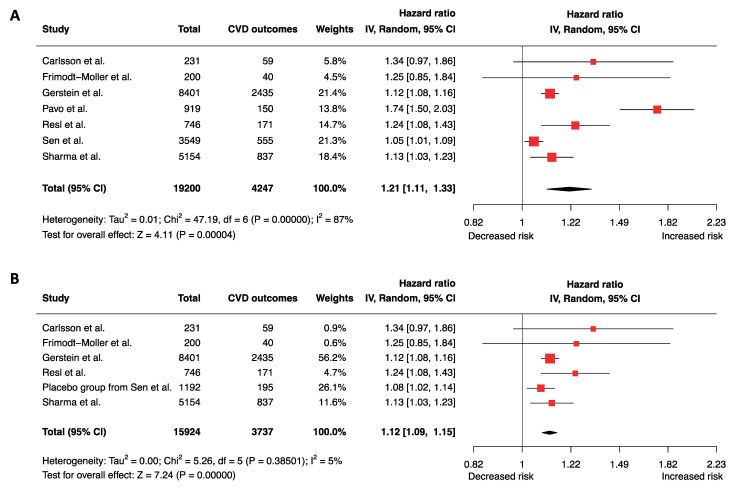
Elevated circulating GDF-15 levels are significantly associated with MACE. Pooled adjusted HRs are shown (**A**) with (HR 1.21, [1.11–1.33], *I*^2^ = 87%) and (**B**) without the unadjusted study by Pavo et al. and subjects who received canagliflozin (*n* = 2357 of 3549) in the study of Sen et al. (HR 1.12, [1.09–1.15], *I*^2^ = 5%).

**Figure 3 biomolecules-12-00934-f003:**
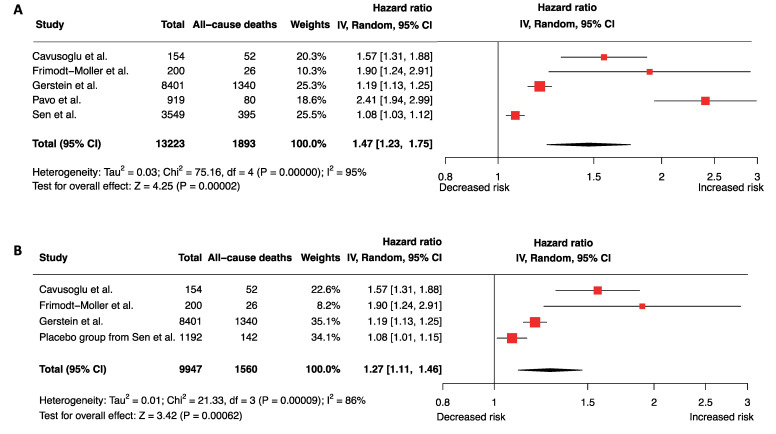
Pooled hazard estimates of all-cause mortality associated with increased circulating GDF-15 levels (**A**) including and (**B**) excluding the unadjusted study by Pavo et al. and canagliflozin received patients (2357 of 3549) from Sen et al. After removal of the unadjusted study, the *I*^2^ decreased 95% to 86%, the pooled effect size reduced from 1.47 to 1.27.

**Table 1 biomolecules-12-00934-t001:** Summary of prospective studies in this meta-analysis.

Study, Year of Publication, Reference no.	Carlsson et al. 2020 [14]	Cavusoglu et al. 2015 [23]	Frimodt-Møller et al. 2018 [13]	Gerstein et al. 2015 [24]	Pavo et al. 2016 [25]	Resl et al. 2016 [26]	Sen et al. 2021 [20]	Sharma et al. 2020 [27]
Country	Sweden	USA	Denmark	Canada	Austria	Austria	Int’l	Int’l
Sample size, *n*	231	154	200	8401	919	746	3549	5154
Study type	Prospective	Prospective	Prospective	Prospective	Prospective	Prospective	Prospective	Prospective
Statistical adjustment (Cox)	Multivariate	Multivariate	Multivariate	Multivariate	Univariate	Multivariate	Multivariate	Multivariate
Median follow-up time, *y*	7.9	5.0	6.1	6.2	5.0	5.0	6.1	1.5
Age, *y*	68	–	59	63.2	62 ^†^	–	62.8	61 ^†^
Male, *n* (%)	169 (73.0)	–	152 (76.0)	5928 (70.6)	511 (55.6)	420 (56.3)	2374 (66.9)	3491(67.7)
BMI, kg/m^2^	30	–	–	–	28.1 ^†^	–	32.7 ^†^	29.5
Smoking, *n* (%)	123 (15.0)	–	59 (29.5)	1050 (12.5)	358 (39.0)	–	–	–
Hypertension, *n* (%)	–	–	–	6638 (79.0)	614 (66.9)	508 (68.0)	–	4291 (83.3)
Heart failure, *n* (%)	–	46 (29.9)	–	–	0 (0)	–	473 (13.3)	1442 (28.0)
Atrial fibrillation, *n* (%)	–	7 (4.5)	–	–	14 (1.6)	–	–	–
Coronary artery disease, *n* (%)	–	130 (84.4)	–	–	105 (11.5)	–	–	–
Myocardial infarction, *n* (%)	–	54 (35.1)	–	–	–	–	4534 (88.0)	–
Duration of diabetes, *y*	–	–	14.7	5.3 ^†^	–	12.0^†^	13.5 ^†^	–
HbA1c, %	7.0	–	–	–	7.1 ^†^	–	8.2 ^†^	8.0
eGFR, mL/min/1.73 m²	70.0	–	91.1	–	73.3 ^†^	72.7 ^†^	77.0 ^†^	70.9
hsTnT ^†^, ng/L	–	–	–	–	8	0.0008	–	9
NT-proBNP ^†^, pg/mL	–	–	–	–	62	67	–	422
GDF-15 ^†^, pg/mL	–	–	1533	–	1391	1474	1774	1246
Medications:								
Aspirin, *n* (%)	–	129 (83.8)	193 (91.5)	–	292 (32.0)	–	–	4683 (90.9)
Statin, *n* (%)	415 (51.0)	98 (63.6)	189 (95.0)	6638 (79.0)	371 (40.4)	317 (42.5)	–	4672 (90.6)
ACEI/ARB, *n* (%)	–	110 (71.4)	–	5793 (69.0)	–	408 (54.7)	–	4247 (82.4)
Beta-blocker, *n* (%)	–	116 (75.3)	–	4526 (53.9)	–	203 (27.2)	–	4240 (82.3)
Any OHA, *n* (%)	152 (65.8)	104 (67.5)	170 (85.0)	–	484 (52.7)	–	–	–
Metformin, *n* (%)	–	55 (35.7)	–	2317 (27.6)	412 (44.8)	339 (45.4)	–	3412 (66.2)
Sulfonylurea, *n* (%)	–	–	–	–	226 (24.8)	196 (26.3)	–	2393 (46.4)
Insulin, *n* (%)	209 (26.0)	42 (27.3)	124 (62.0)	–	597 (65.0)	508 (68.0)	–	1540 (29.9)

^†^ Continuous variables are reported as mean or median. ACEI/ARB: angiotensin converting enzyme inhibitor/angiotensin receptor blocker; BMI: body mass index; eGFR: estimated glomerular filtration rate; GDF-15: growth differentiation factor-15; hsTnT: high-sensitivity troponin T; Int’l: international; NT-proBNP: N-terminal prohormone of B-type natriuretic peptide; OHA: oral hypoglycemic agent; –: not applicable.

**Table 2 biomolecules-12-00934-t002:** Definition of study endpoints, MACE, and adjusted confounders in the included studies.

Study (Year)	Endpoint	Definition of MACE	Adjusted Confounders	Ref.
Carlsson et al. (2020)	MACE	(1) Fatal or nonfatal MI or stroke	Age, sex, frailty, microalbuminuria, renal function, CVD at baseline, smoking, LDL, and SBP	[14]
	All-cause death	–	–	
Cavusoglu et al. (2015)	MACE	–	–	[23]
	All-cause death	–	Age, HF or MI at presentation, extent of angiographic CAD, eGFR, metformin use, TZD use, and ST2	
Frimodt-Møller et al. (2018)	MACE	(1) Incidence of CV death, nonfatal MI, stroke, ischemic CVD, and HF	Age, sex, smoking status, systolic BP, LDL, HbA1c, plasma creatinine, and urinary albumin excretion rate	[13]
	All-cause death	–	Age, sex, smoking status, systolic BP, LDL, HbA1c, plasma creatinine, and urinary albumin excretion rate	
Gerstein et al. (2015)	MACE	(1) Composite of MI, stroke, or CV death (2) Outcome 1 plus HHF or revascularization	Age, sex, smoking status, prior DM, HT and CV events, LDL/HDL, albuminuria, and levels of serum creatinine, NT-proBNP, chromogranin A, Ang-2, GSTA, apolipoprotein B and tissue inhibitor of metalloproteinase 1	[24]
	All-cause death	–	Age, sex, smoking status, prior DM, HT and CV event, LDL/HDL, albuminuria, and levels of serum creatinine, NT-proBNP, chromogranin A, Ang-2, GSTA, trefoil factor 3, α-2-macroglobulin, tenascin, selenoprotein P, macrophage derived chemokine, YKL-40 and IGF binding protein 2	
Pavo et al. (2016)	MACE	(1) CV death	–	[25]
	All-cause death	–	–	
Resl et al. (2016)	MACE	(1) Composite endpoint of unplanned hospitalization for CV events or CV death secondary to MI, decompensated HF, worsening arrhythmia, critical limb ischemia or venous thromboembolism	Age, sex, and log-transformed duration of DM, BP, eGFR, LDL, total cholesterol, HbA1c, urinary albumin excretion and NT-proBNP	[26]
	All-cause death	–	–	
Sen et al. (2021)	MACE	(1) Nonfatal MI, nonfatal stroke, or CV death	Age, sex, race, and randomized treatment assignment (canagliflozin or placebo) *, history of CVD, HbA1c, systolic and diastolic BP, BMI, LDL cholesterol, eGFR and UACR	[20]
	All-cause death	–	Age, sex, treatment assignment, UACR, eGFR, and CVD history	
Sharma et al. (2020)	MACE	(1) Composite of CV death, HHF, initiation of loop diuretics, or NT-proBNP elevation (2) Composite of CV death or HHF	Age, sex, smoking status, systolic BP, history of HF, duration of DM, prior MI, HT, hyperlipidemia, and eGFR	[27]
	All-cause death	–	–	

Ang-2: angiopoietin-2; BMI: body mass index; BP: blood pressure; DM: diabetes mellitus; CAD: coronary artery disease; CVD: cardiovascular disease; DM: diabetes mellitus; eGFR: estimated glomerular filtration rate; GSTA: glutathione-S-transferase A; HDL: high-density lipoprotein; HF: heart failure; HHF: hospitalization for heart failure; HT: hypertension; IGF: insulin-like growth factor; LDL: low-density lipoprotein; MACE: major adverse cardiovascular event; MI: myocardial infarction; Ref.: reference number; ST2: suppression of tumorigenicity 2; TZD: thiazolidinedione; UACR: urine albumin-to-creatinine ratio; –: not applicable. * Note the absence of treatment effect on GDF-15 levels reported in the randomized controlled trial, CANVAS.3.2. Elevated GDF-15 and Risks of Future MACE.

## Data Availability

Data collected for this study are available upon request. Queries should be directed to the corresponding author: e.fung@cuhk.edu.hk.

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
