# Peer review of "Major Adverse Cardiovascular Events and Mortality Prediction by Circulating GDF-15 in Patients with Type 2 Diabetes: A Systematic Review and Meta-Analysis"

_biomolecules, 2022, doi:10.3390/biom12070934_

Round 1

Reviewer 1 Report

I read with great interest this review article. It is quite well written, well organized, structured, it includes an informative flow diagram, and also the statistics appear to be very well conducted. 

As minor issue, I think that some additional information about GDF-15 could be of help in the introduction section (including origin, cell type expressing and secreting it, cells target, etc). 

“The effects of circulating GDF-15 on adverse outcomes in patients with HF or coronary artery disease…”; are there experimental models exploring the way by the which GDF-15 could exert detrimental effects? Could you speculate that GDF-15 elevation is not a causative/risk factor for CVD, but the mirror of the increased CVD risk in a homeostatic model?

“And the therapeutic effects of antidiabetic medications; a reasonable question to be raised could be: What about the role of canagliflozin? But the authors already tried to make sense to the results reported about as available so far (however, please check if any further useful data have been more recently reported to date). Are there any further available data about GLP-1 and GDF-15? I think that including these report(s) could be of help in increasing the knowledge on GDF-15 role in T2DM. 

Since authors have considered them, are they able to report the comorbidity rates of the included T2DM patients (not only hypertension, but also dyslipidemia, chronic kidney disease, etc), also as pooled?

Author Response

We thank Reviewer 1 for the encouraging and very helpful feedback.

Please find below our point-by-point response to the questions.

1. As minor issue, I think that some additional information about GDF-15 could be of help in the introduction section (including origin, cell type expressing and secreting it, cells target, etc). 

Thanks for the suggestion.  We have added ~5 lines (lines 66-71) of information to Introduction to give a more comprehensive portrayal of its biology.

2. “The effects of circulating GDF-15 on adverse outcomes in patients with HF or coronary artery disease…”; are there experimental models exploring the way by the which GDF-15 could exert detrimental effects?

GDF-15 may exert detrimental effects in at least two ways: 1) GDF-15 binding to GFRAL in the hindbrain (area postrema and nucleus tractus solitarius) induces anorexia, leading to weight loss and cachexia; 2) GDF-15 acting on skeletal muscle can induce loss of muscle bulk (sarcopenia). In 2017, four extraordinary papers were published in Nature and Nature Medicine that demonstrated the critical roles of GDF-15/GFRAL in the maintenance of metabolic homeostasis. Prolonged or chronic increases in GDF-15 can act centrally to induce a state of muscle wasting and cachexia, which is known to be associated with increased mortality. This is being explored by pharmaceutical companies to treat obesity. In peripheral adipose tissue, GDF-15 can play a self-regulatory autocrine role as adipocytes can both secrete and respond to GDF-15.

3. Could you speculate that GDF-15 elevation is not a causative/risk factor for CVD, but the mirror of the increased CVD risk in a homeostatic model?

Great question.  GDF-15 elevation can be attributable to both genetic etiology and other complex lifestyle factors (e.g. smoking, diet, physical activity) in the development of CVD. Data on this topic are very scarce. Several studies have found that single nucleotide polymorphisms (SNPs) in the GDF15 gene (at chromosome 19p13.11) were associated with variable levels of circulating GDF-15 that were also influenced by cardiometabolic risk factors (Ho J et al. Clin Chem 2012;58:1582-1591). A meta-analysis found that nine SNPs in the GDF15 gene explained 21.47% of the variance in blood GDF-15 concentration (Jiang J et al. Front Genet 2018;9:97). However, a recent Mendelian randomization study found no evidence of a causal association between GDF-15 SNPs and the incidence of incidence of stroke, heart failure and non-ischemic cardiomyopathy (Wang Z et al. BMC Cardiovasc Disord 2020 Oct 28;20:462). We have added a new paragraph in Discussion to provide additional details on the above (lines 264-276).

4. And the therapeutic effects of antidiabetic medications; a reasonable question to be raised could be: What about the role of canagliflozin? But the authors already tried to make sense to the results reported about as available so far (however, please check if any further useful data have been more recently reported to date). Are there any further available data about GLP-1 and GDF-15? I think that including these report(s) could be of help in increasing the knowledge on GDF-15 role in T2DM.

At present, it may be premature to conclude the effects of SGLT2 inhibition on GDF-15 levels. More data from studies are needed. Regarding GLP-1 receptor agonist and GDF-15 levels, there are at least two studies that have demonstrated no significant change in response to treatment with the antidiabetic drug: Sharm A et al. ESC Heart Fail 2021;8:2608-2616 (PMID 34061470), and Valenzuela-Vallejo L et al. Metabolism 2022 Jun 11;133:155237 (PMID 35700837). 

5. Since authors have considered them, are they able to report the comorbidity rates of the included T2DM patients (not only hypertension, but also dyslipidemia, chronic kidney disease, etc), also as pooled?

Thanks for the suggestion.  We newly added heart failure, atrial fibrillation, coronary artery disease and myocardial infarction to Table 1. As only 1 study reported on dyslipidemia, we did not include that in the table. We also did not add chronic kidney disease as the mean eGFR is already shown and provides an estimate for the kidney function of the study population.

Reviewer 2 Report

Dear Editor,

I really appreciate this metanalysis from Xie et al. The authors well addressed this important issue in the field of biomarkers in cardiovascular diseases. The manuscript is well written and the ENglish of the paper is fluent. THe figures are ok and clear. 

Author Response

We thank Reviewer 2 for reviewing this manuscript.